# Design and Modeling of Ultra-Compact Wideband Implantable Antenna for Wireless ISM Band

**DOI:** 10.3390/bioengineering10020216

**Published:** 2023-02-06

**Authors:** Ahmed Z. A. Zaki, Ehab K. I. Hamad, Tamer Gaber Abouelnaga, Hala A. Elsadek, Sherif A. Khaleel, Ahmed Jamal Abdullah Al-Gburi, Zahriladha Zakaria

**Affiliations:** 1Communication Department, Modern Academy for Engineering and Technology, Cairo 11571, Egypt; 2Electrical Engineering Department, Faculty of Engineering, Aswan University, Aswan 81542, Egypt; 3Microstrip Circuits Department, Electronics Research Institute (ERI), Elnozha, Cairo 11843, Egypt; 4Higher Institute of Engineering and Technology, Kafr El-Shiekh 33511, Egypt; 5College of Engineering and Technology, Arab Academy for Science Technology and Maritime Transport, Aswan 81511, Egypt; 6Center for Telecommunication Research & Innovation (CeTRI), Fakulti Kejuruteraan Elektronik dan Kejuruteraan Komputer (FKEKK), Universiti Teknikal Malaysia Melaka (UTeM), Durian Tungal, Malacca 76100, Malaysia

**Keywords:** antenna for biomedical applications, implanted antenna, medical implant communication service band (MICS), medical monitoring, wireless body area network (WBAN)

## Abstract

This paper proposes a wideband ultra-compact implantable antenna for a wireless body area network (WBAN). The proposed patch antenna works in the industrial, scientific, and medical (ISM) bands. The proposed patch antenna with an ultra-compact size (5 × 5 × 0.26 mm^3^) was designed with 29% wide bandwidth (about 670 MHz). This wide bandwidth makes the antenna unaffected by implantation in different human body parts. The miniaturization process passed many steps by adding many slots with different shapes in the radiating element as well as in the ground plane. A 50 Ω coaxial feeding excites the antenna to maintain matching and low power loss. The specific absorption rate (SAR) was calculated for health considerations. The result was within the standard limits of IEEE organizations and the International Commission on Non-Ionizing Radiation Protection (ICNRP). The antenna was tested in tissues with multiple layers (up to seven layers) and at various depths (up to 29 mm). The link margin was calculated, and the proposed antenna enables 100 Kbps of data to be transferred over a distance of 20 m and approximately 1 Mbps over a distance of 7 m. The proposed antenna was fabricated and tested. The measured S_11_ parameters and the simulated results using the Computer Simulation Technology (CST Studio) simulator were in good agreement.

## 1. Introduction

Non-communicable diseases (NCDs) such as heart disease, cancer, chronic respiratory diseases, and diabetes are among the leading causes of death worldwide. Approximately 60% of all deaths annually around the world are due to non-communicable diseases [1]. By 2025, the number of people over the age of 60 is expected to rise to 1.2 billion (approximately 15% of the world’s population) [2]. This will lead to an increase in the number of people with non-communicable diseases, which will prompt the world to find new solutions to detect NCDs early to contribute to the lower cost of treatment [3]. Collecting data from remote patients will relieve pressure on healthcare systems due to the early detection of any change in the body’s vital functions by placing sensors implanted inside the body or worn by humans. Wearable/implantable sensors can be distributed around the body to collect data and are linked together via a wireless link called the wireless body area network (WBAN) [4]. The function of these sensors is to monitor the vital functions of the body around the clock. There is a rapid increase in the rate of WBAN usage, with an increasing number of WBAN-based devices in use. Both wearable/implantable devices reached nearly 18 million units in 2017, and the number is expected to be doubled by 2025 [3,4]. As shown in Figure 1, the configuration of an implantable/wearable WBAN consists of wearable sensors and a set of implantable sensors. Implanted WBAN sensors are part of an implanted medical device (IMD). Usually, the IMD consists of an ultra-miniaturized antenna, data/power management unit, biosensor, and battery [5,6,7].

The implanted devices collect data and send it to a wearable data display device or a remote-controlled unit located outside the sensor and vice versa. Implanted devices can communicate with each other, as found in glucose sensors and insulin injection devices [8]. Some specifications must be considered in the IMD devices, such as their small size [9,10], which does not exceed 1 cm^3^ [11], low output power to avoid any health concerns to the patient [12], and low power consumption to preserve the battery if possible [13]. One of the biggest challenges that attracted the attention of researchers is to reduce the size of the IMD to allow its manufacture in the form of a capsule [14,15], which required reducing the size of all IMD components. Therefore, the design of an ultra-miniaturized antenna will contribute to the accomplishment of this goal, and a microstrip patch antenna has recently gained researchers’ attention [16].

The antenna of the IMD must operate on the following frequency bands: medical implant communication service (MICS) (402–405 MHz) and industrial scientific and medical (ISM) bands (902–928 MHz, 2.4–2.4835 GHz, and 5.725–5.875 GHz) [17,18]. The MICS band is most commonly used, but it has a narrow band and a low data rate. Recently, ISM bands have been preferred because of their wide band, high data rate, and small antenna size [19,20]. 

One of the essential requirements of the implanted antenna is the stability of performance within the layers of the human body (human tissue). The performance of the implanted antenna directly depends on the dielectric properties of the surrounding medium [21,22]. The layers of the human body are naturally heterogeneous (each layer has different dielectric properties). Changing the properties of surrounding human tissues affects the resonance frequency. It can cause an uncontrolled shift, which is called the detuning effect, so the wideband antenna is demanded to mitigate this effect [23,24]. In [24], the authors were interested in studying the detuning of the antennas implanted inside the human body, so three stacked layers of the antenna were designed with a volume of 384 mm^3^. The antenna operated at 403 MHz with 66 MHz bandwidth. The results showed that a large frequency shifting occurred when the antenna was implanted in some parts of the body, such as the eye and the ear. In [25], a robust parameter design was used to reduce detuning effects with an antenna volume of 203 mm^3^. An ultra-wideband antenna with a volume of 28.85 mm^3^ was designed to mitigate detuning effects, which was reported in [26] with a maximum gain of −30 dB. There is still a significant challenge in designing this type of antenna to overcome the detuning effect and achieve balance between the antenna size and acceptable gain rate. Some research suggests different methods for designing a wideband antenna for biomedical applications, such as increasing the substrate thickness [27], a meandered strip antenna [28], and different-shaped slots in patches and the ground [29,30]. 

This work proposes a new wideband ultra-compact implantable antenna (WUCIA) for biomedical applications resonating over the ISM band of 2.45 GHz with a wideband of about 29%. The proposed ultra-compact antenna has a 6.5 mm^3^ volume, achieved by adding different-shaped slots in the radiator and a partial ground. The wide bandwidth allows the antenna to overcome the frequency deviation (detuning effect) as a result of implantation in different parts of the body. The antenna was tested in tissues with multiple heterogeneous layers (up to seven layers) such as skin, kidney, muscle, liver, and brain at various depths (up to 29 mm), with a maximum achieved gain −24 dBi (implanted under the skin layer). The SAR was examined using CST studio to assure patient safety, and it was discovered to be in compliance with IEEE and ICNRP safety criteria for various implanted organs. The link budget was computed to assess the range of data transfer that could be covered. The proposed antenna achieved up to 20 m at 100 kbps data transmission and 6.5 m at 1 Mbps data transmission. The proposed antenna was able to achieve a great balance between the antenna size (6.5 mm^3^), acceptable gain (−24 dBi), and large bandwidth (670 MHz) to overcome the detuning effect.

## 2. Background

### 2.1. Size Reduction Techniques

Recently, reducing the size of the antenna implanted inside the human body is one of the challenges facing researchers to make it suitable for use in IMD devices. Some techniques can help reduce the size of the antenna, such as adding pins [31], which will contribute to doubling the effective size of the antenna and reducing its physical size. Additionally, adding slots in the ground leads to an increase in capacitance [32]. Similarly, adding slots in the antenna radiator leads to an increase in the length of the current path [33]. Adding circular slots such as split-ring resonators (SRR) [34] or using a superstrate assists in the miniaturization [35]. Additionally, using two or more radiating patches as the stack [36] can greatly help in miniaturization. The medium surrounding the implanted antenna can help in miniaturization. For example, at a frequency of 2.45 GHz, the wavelength in the free space is about 122 mm, while it is only about 19 mm when the antenna is implanted inside the skin layer with *ε_r_* = 38, as recorded in Table 1.

### 2.2. Properties of Human Tissues

After the antenna is implanted inside the human body, human tissue becomes the medium surrounding the implanted antenna. The performance of the implanted antenna is affected by any change in the electrical properties of this medium. Additionally, the properties of the medium can change with the age, weight, or gender of the patient. Due to the heterogeneity of human tissue layers, the implanted antenna faces many challenges to wireless transmission.

Human tissue consists of several layers (skin, fat, and muscle). The arrangement of the layers varies according to where the antenna is implanted. Each of these layers has different electrical properties that change with frequency. The dielectric constant (*ε*) and conductivity (*σ*) of different layers of the human tissues at 2.45 GHz are mentioned in Table 1. The dielectric constant directly affects the wavelength of the implanted antenna. The wavelength of the antenna in free space is calculated as:
(1)$$\lambda_{o}=\frac{C}{f}$$

The wavelength of the implanted antenna can be calculated [37]:(2)$$\lambda_{g}=\frac{\lambda_{o}}{\sqrt{\varepsilon_{r}}}$$

So, the medium surrounding the implanted antenna can help in the miniaturization process of the antenna [35].

**Table 1 bioengineering-10-00216-t001:** The dielectric properties of the different layers of the human body at 2.45 GHz [38,39].

Tissue Type	Dielectric Constant (*ε_r_*)	Conductivity *σ* (S/m)
Skin	38	1.44
Fat	5.28	0.1
Muscle	52.7	1.74
Kidney	52.9	2.37
Liver	43	1.69
Brain Layers		
Bone	11.41	0.394
CSF	66.3	3.46
Dura	42.1	1.67
Grey Matter	49	1.81
White Matter	36.2	1.21

## 3. Antenna Design and Simulations

### 3.1. Proposed Antenna Configuration

The configuration with the overall dimensions of the proposed wideband ultra-compact implanted antenna (WUCIA) is illustrated in Figure 1. The proposed WUCIA consisted of a meandered radiator patch with different-shaped slots and a partial ground as shown in Figure 2a,b. A Roger RO3003 with a dielectric constant equal to 3 was used as a substrate and superstrate with dimensions 5 × 5 mm^2^ and a thickness of 0.13 mm.

The superstrate assists in the miniaturization process as well as prevents direct contact between the radiating element and human tissues. The antenna was covered with ceramic alumina (Al_2_O_3_) of 0.02 mm thickness with a dielectric constant of 9.8 and a loss tangent of 0.008 to guarantee patient safety and avoid a short circuit [40]. A 50 Ω coaxial cable was used to feed the antenna. The excitation was placed at *x* = −1.75 mm and *y* = −1.75 mm from the center. The optimized parameters of the proposed WUCIA are tabulated in Table 2.

### 3.2. Size Reduction Steps

Figure 3 describes the antenna design steps and the corresponding current distributions. To obtain the desired resonant frequency, horizontal and vertical slots were added to the square patch to obtain a meandered antenna radiating surface, which helps to extend the current path and helps in the antenna miniaturization process. The design started in the first step with a conventional square patch antenna based on Balanis equations [27] with two vertical slots (S6 & S9), and the result was a very weak resonance of less than −5 dB, as shown in Figure 4 (step 1). In the second step, two horizontal slots (S1 & S12) were added to lengthen the current path, and the result was a 2.8 GHz resonant frequency. In the third step, more than one slot was added, resulting in the resonant frequency being decreased to 2 GHz. Step four was to improve the impedance matching, hence an additional slot was added near the feeding point, as well as a partial ground to improve the bandwidth.

### 3.3. Analysis and Discussion

Usually, the antenna is implanted in the muscle layer (under the fat layer) [41,42,43], so the proposed WUMA was designed in the muscle layer using CST studio package. Human body tissue consists of skin, fat, and muscle [44]. The electrical properties of each layer are mentioned in Table 1. The proposed WUMA was tested in the three-layer phantom to imitate typical human tissue, as shown in Figure 5a. The implanted antenna was placed at a 2 mm distance from the surface of the muscle layer and at a 9 mm distance from the air. As illustrated in Figure 5b, the proposed WUCIA operates at 2.29 GHz with a −10 dB impedance bandwidth of 22%. The proposed WUCIA has a gain of −26 dBi, which is a negative gain due to the losses caused by the surrounding human lossy tissue.

### 3.4. Sensitivity Analysis and Parametric Study

As mentioned before, the implanted antenna is affected by the electrical properties of the surrounding medium, so the antenna was tested in different mediums to evaluate its performance in a precise manner. For this purpose, the values of *ε_r_* and *σ* of the antenna-surrounding tissues (muscle) changed from 60% to 120% from its original values at 2.45 GHz, i.e, the *ε_r_* of the surrounding tissues varied from 31 to 64 and the *σ* of muscle changed from 1 to 2.1, as shown in Figure 6. As shown in Figure 6a, the value of the dielectric constant of the tissues changed from 60% to 120% of its value at 2.45 GHz, and the resonant frequency shifted towards a lower frequency as *ε_r_* increases, and vice versa; this agrees with Equation (3) that there is an inverse relationship between *f_r_* and *ε_r_* [45].
(3)$$f_{r}=\frac{c}{\lambda_{g}\surd \varepsilon_{\mathrm{eff}}}$$

As shown in Figure 6b, *σ* had a negligible effect on the resonant frequency. According to Equation (3), *σ* does not directly affect the frequency. The variations in *σ* as well as in *ε_r_* of the tissues could affect the impedance matching of the antenna [46], as shown in Figure 6a,b. Additionally, the variation in the values of *ε_r_* and *σ* could affect the loss tangent (tan δ), which is calculated by:
(4)$$\tan\;\delta =\frac{\sigma }{\varepsilon_{0}\varepsilon_{r}\omega }$$

Some parameters had to be studied due to the different natures of human bodies. For example, the thickness of the fat layer varies from person to person and from one area to another area within the body. For example, the fat layer may reach a thickness of 30 mm in the abdominal area [47]. Therefore, our proposed implanted antenna was tested at different fat layer thicknesses, as shown in Figure 7.

The antenna showed good stability in the reflection coefficient response despite changing the thickness of the fat layer from 4 up to 25 mm, as illustrated in Figure 7a. The antenna gain changed, as depicted in Figure 7b, but it was still in the range that did not impede the data transmission. Again, the implanted antenna was tested at different implantation depths from 4 up to 30 mm from the surface of the muscle tissue. The result was quite stable in terms of S_11_, as illustrated in Figure 8a. Additionally, the gain changed a little, but it was within the acceptable range, which did not affect the transmission of data, as shown in Figure 8b.

A parametric study of some parameters of the antenna is essential for assisting in selecting the optimal dimensions of the antenna and determining the main parameters affecting the antenna performance. The parametric study was carried out on the human model built on Microwave Studio CST, as mentioned before in Figure 5, to test the influence of these parameters on the S_11_ of the proposed antenna. The impact of the variations of the antenna’s parameters S10, S6, S12, and partial ground length (G) on the reflection coefficient (S_11_) is illustrated in Figure 9. Some of the parameters used directly affected the resonant frequency, such as “S10” and “S12”, while other parameters such as “S6” and “G” affected the bandwidth and matching impedance of the antenna.

### 3.5. Model Integrity Analysis

To evaluate the antenna’s performance and its response to different environmental surroundings and to consider a more realistic scenario, the antenna was implanted in different areas of the human body model, such as the liver, kidney, brain, and under the skin layer, as demonstrated in Figure 10. To appropriately implant the antenna in the desired part of the body, we should know the anatomy of the layers of the body at each area where the implanted antenna is going to be placed. For example, when the antenna was implanted inside the human kidneys, it was found that the kidneys were located on the posterior abdominal wall consisting of a musculoskeletal structure [47]. So, the body tissue at the kidney consists of skin, fat, muscle, and kidney, in that order. The proposed antenna was placed 2 mm from the kidney layer surface and 29 mm from the air, as shown in Figure 11a.

In the case of implanting the antenna in the liver, the simulation model consisted of four layers (skin, fat, muscle, and liver) [47]. The proposed antenna was placed 2 mm from the liver layer surface and 29 mm from the air, as shown in Figure 11b. The brain model consisted of seven layers (skin, fat, bone, dura, cerebrospinal fluid (CSF), and grey and white matter); the layer dimensions and implanting depth were mentioned in [48]. The proposed antenna was placed 1 mm from the dura layer surface and 11 mm from the air, as shown in Figure 11c,d. The human tissue consisted of three layers: skin, fat, and muscle [44]. The proposed antenna was placed 2 mm from the skin layer surface. The simulation model of the human tissue was built on the CST EM simulator. The thickness of each layer was chosen as an average value because the thickness of the layers differs from one person to another and depends on where the antenna is implanted in the human body.

The simulated S_11_ parameter, gain, total efficiency, and input impedance of the proposed WUCIA for different implementation scenarios are illustrated in Figure 12. The proposed antenna for different scenarios attained a good response in terms of the reflection coefficient (S_11_), as depicted in Figure 12a. The achieved −10 dB impedance bandwidths were 22% (510 MHz), 29% (670 MHz), 24.8% (610 MHz), 26% (654 MHz), and 20% (520 MHz). The accomplished gains were −26 dBi, −30 dBi, −31 dBi, −25 dBi, and −24 dBi for muscle, kidney, liver, brain, and skin, respectively.

The simulated E- and H-*plane* radiation pattern was calculated at 2.45 GHz of the UMPA in the different implementation scenarios that are illustrated in Figure 13, where the antenna was placed in the *x*–*y* plane. The shape of the radiation pattern depends on the implantation sites. The radiation patterns are almost omni-directional when the antenna is implanted in muscle, kidney, and skin, while the E- and H-planes resemble a circle in the case of the antenna implanted in the liver and brain. The achieved radiation pattern was accepted as a good candidate for biomedical applications in all scenarios.

## 4. Specific Absorption Rate Calculations

The specific absorption rate (*SAR*) is the rate of RF energy absorbed by the body from the antenna. To ensure no harm can be caused by the radiated electromagnetic field of the implanted antenna to patients wearing IMD, the *SAR* should not exceed the standard limits set by the approved organizations. The ICNRP limits the peak average *SAR* for 1 g of tissue as 2 W/Kg, and the IEEE C95.1-1999 limit for 10 g of tissue is 1.6 W/kg [49]. The *SAR* can be calculated by:
(5)$$SAR=\sigma {\left|E\right|}^{2}/\rho_{den}$$
where *σ* is the conductivity of human tissue, *E* is the intensity of the electric field, and $\rho_{den}$ is the density of human tissue. The maximum *SAR* and the maximum allowed input power for 1 g/10 g body tissue are tabulated in Table 3. The *SAR* was determined in different body sites (muscle, kidney, liver, brain, and skin). The input power to the antenna was set as 1 W at 2.45 GHz. As shown in Table 3, the *SAR* values depend on where the antenna is implanted due to the difference in the electrical properties of such medium.

## 5. Link Budget

The evaluation of the telemetry range between implanted and external devices is an important issue. The implanted devices transfer the biological information of the patient to an external monitor device. The external device may be located several meters from the patient’s body, and the implanted antenna should be able to communicate with the external device, so the link margin (*LM*) needs to be calculated [41].
(6)$$LM\;\left(\mathrm{dB}\right)=CL\;\left(\mathrm{dB}\right)-RL\;\left(\mathrm{dB}\right)$$

To guarantee a stable link, the current link (*CL*) must be higher than the required link (*RL*), i.e., *LM* should be greater than zero dB (+ve value). The current link (*CL*) is given by:
(7)$$CL\;\left(\mathrm{dB}\right)=P_{t}+G_{t}-L_{f}-L_{a}+G_{r}-2L_{feed}-N_{o}$$
where *P_t_* is the *T_x_* power, *L_feed_* is the feeding loss, *G_t_* is the gain of the transmitter’s antenna, *L_f_* is the free space propagation loss, *L_a_* is the air propagation loss, *G_r_* is the gain of the receiver’s antenna, and *N_o_* is the noise power density. The current link is significantly affected by the distance between the implanted device and the external device because *L_f_* depends on this distance (*d*):
(8)$$L_{f}\;\left(\mathrm{dB}\right)=\frac{2\pi d}{\lambda }$$

The required link (*RL*) is given by:
(9)$$RL\;\left(\mathrm{dB}\right)=E_{b}/N_{o}+10\;\log\;\left(B_{r}\right)-G_{c}+G_{d}\left(\mathrm{dB}\right)=\frac{2\pi d}{\lambda }$$

*E_b_/N_o_* is the normalized signal-to-noise ratio, *B_r_* is the bit rate, *G_c_* is the coding gain, and *G_d_* is the fixing deterioration. The key parameters of this calculation are tabulated in Table 4 [50].

In our calculations, the maximum value for polarization losses was considered due to the use of a linear polarized antenna that can change its orientation in response to the patient’s motion. Assuming a linear dipole antenna with a gain of 2.15 dB is placed in the receiver, the input power of the antenna is restricted to 25 μW by the European Research Council [31]. As demonstrated in Figure 14, the *LM* of the implanted antenna was calculated in different scenarios. There was an inverse relationship between *B_r_* and the distance between the implanted antenna and the external monitor. The implanted antenna could send and receive up to 20 m with stability at 100 Kbps. This distance decreased to approximately 6.5 m at 1 Mbps data transfer, which allows the antenna implanted in the endoscope to send images captured inside the body.

## 6. Fabrication and Measurements

The measurement and fabrication of the designed antenna are essential steps in verifying the validation of the numerical calculations. So, the proposed antenna was fabricated using the photolithography technique and measured using a vector network analyzer (VNA Rohde & Schwarz “ZVB20”) in the Electronics Research Institute’s Labs. The images of the prototype proposed antenna are displayed in Figure 15 (top and bottom views) as well as the antenna joint of the connector used. The antenna was tested and measured using fresh beef steak meat, kidney, and liver because it is difficult to experiment using the human body. The dielectric property of beef muscle is *ε_r_* = 53.69, fat is *ε_r_* = 3.6 at 2.45 GHz [51,52,53], kidney *ε_r_* is 51, and liver *ε_r_* is 46.6 [54,55]. The piece of meat was selected as closely as possible to that used in the simulations in terms of skin, fat, muscle, liver, and kidney dimensions.

Photos for the experimental setup are displayed in Figure 16. The muscle model consisted of three layers (skin, fat, muscle), as shown in Figure 16a, the liver model consisted of four layers (skin, fat, muscle, liver), as shown in Figure 16b, and the kidney model consisted of four layers (skin, fat, muscle, kidney), as shown in Figure 16c. The simulated and measured reflection coefficients of the implanted antenna in beef muscle, liver, and kidney are illustrated in Figure 17. The measurements showed a good agreement between the simulated and measured results. Additionally, the antenna had good bandwidth stability while changing the medium. A minor shift in the resonant frequency as well as slight degradation in the matching S_11_ values may be caused by unexpected fabrication tolerance or soldering roughness.

Comparison of proposed antenna and similar prior studies in recent years in terms of frequency, volume, gain, BW, SAR, implementation scenario, depth, and conditions have been tabulated in Table 5.

## 7. Conclusions

This article discussed the design of a wideband ultra-compact antenna for an ISM band wireless body area network for health monitoring with an overall volume of 5 × 5 × 0.26 mm^3^. The proposed WUCIA was optimized to operate over the ISM band with a wide bandwidth of about 670 MHz at 2.45 GHz. The proposed WUCIA was designed mainly to address the detuning effect that may be caused by the heterogeneity of human tissue or changes in tissue properties with aging, and was embedded with IMD circuitry. The optimization of the designed antenna parameters was performed by parametric analysis using Microwave Studio CST. The link budget was computed to assess the range of data transfer that could be covered. The proposed antenna achieved up to 20 m at 100 kbps data transmission and 6.5 m at 1 Mbps data transmission. The SAR was calculated and compared to standards to be within safety limits. To verify and confirm the numerical calculations and simulated result, the designed antenna was fabricated and tested. Due to the difficulty of testing it using the human body, the antenna was tested and measured in some organs of cows, due to the closeness of their characteristics to the organs of the human body, such as muscles, kidneys, and liver, and the result was good. The measured S_11_ parameters and the simulated results using the Computer Simulation Technology (CST Studio) simulator were in good agreement. The measured bandwidth was wide enough to cover the whole ISM band and overcome any unexpected shift in frequency due to losses of human tissue materials. The achieved gains were −26 dB, −30 dB, −31 dB, −25 dB, and −24 dB for muscle, kidney, liver, brain, and skin, respectively. The gain was acceptable for such an antenna size, but needs more improvement in the future. The proposed WUCIA demonstrated a strong potential for use as an implanted antenna for WBAN applications in the ISM band for health monitoring.

## Figures and Tables

**Figure 1 bioengineering-10-00216-f001:**
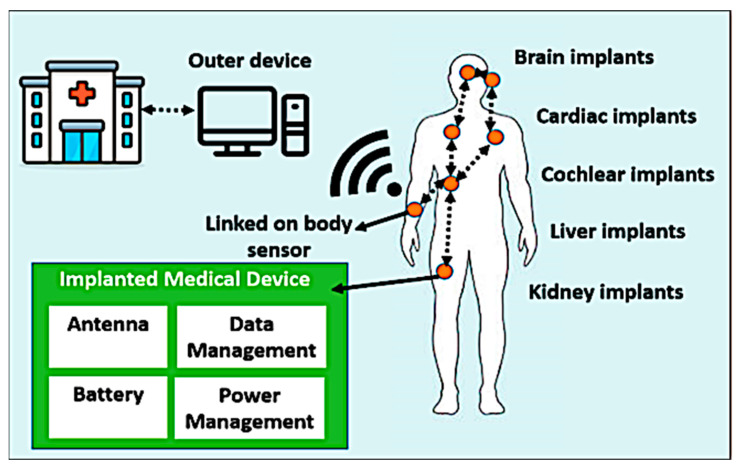
Implanted WBAN for healthcare monitoring.

**Figure 2 bioengineering-10-00216-f002:**
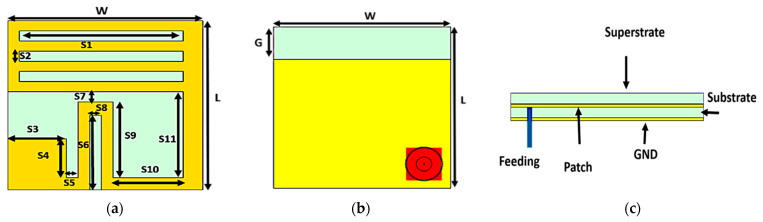
Configuration of the proposed WUCIA: (**a**) top view; (**b**) bottom view; (**c**) side view.

**Figure 3 bioengineering-10-00216-f003:**
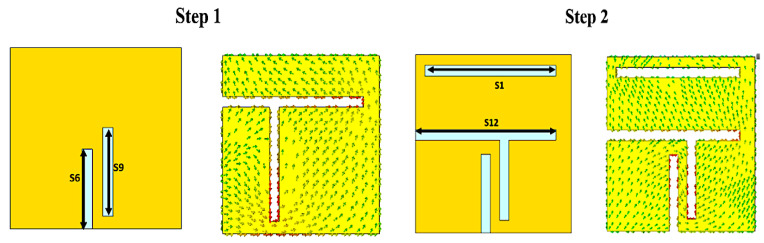
Design steps of the proposed UMPA along with the respective current distributions. Step 1: Conventional square patch with two vertical slots (S6 & S9). Step 2: Two horizontal slots (S1 & S12) were added to lengthen the current path. Step 3: More than one slot was added to shift the frequency. Step 4: An additional slot was added near the feeding to improve matching.

**Figure 4 bioengineering-10-00216-f004:**
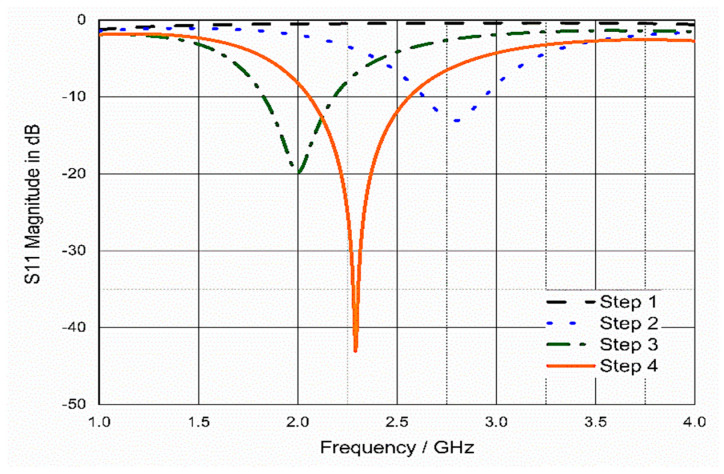
S_11_ of the four design steps of WUCIA.

**Figure 5 bioengineering-10-00216-f005:**
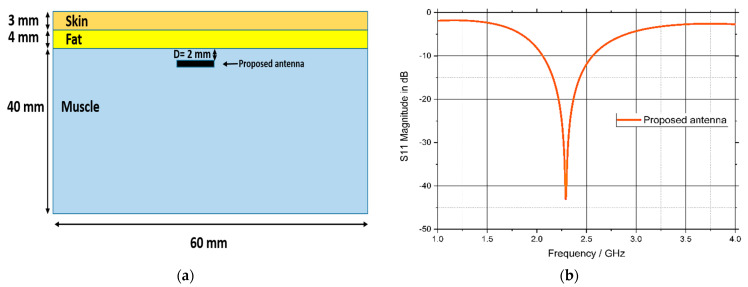
(**a**) Biological human body tissue model in CST Microwave Studio; (**b**) Reflection coefficient of the proposed WUCIA.

**Figure 6 bioengineering-10-00216-f006:**
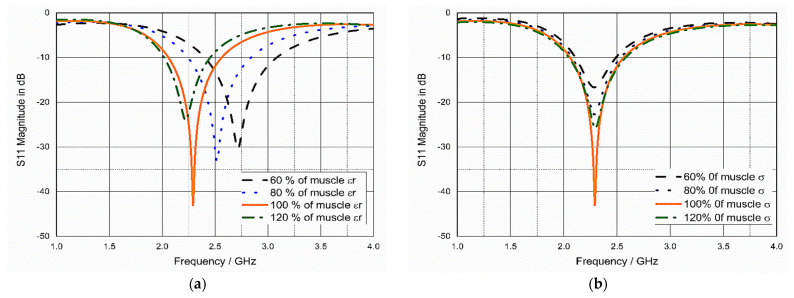
Sensitivity analysis: (**a**) S_11_ in terms of relative permittivity of muscle, *εr*, which changed from 60% to 120% from its values at 2.45 GHz, and (**b**) S_11_ in terms of conductivity of muscle, *σ*, which changed from 60% to 120% from its values at 2.45 GHz.

**Figure 7 bioengineering-10-00216-f007:**
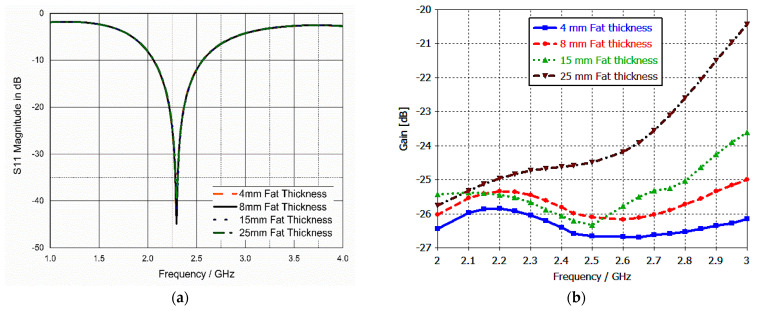
(**a**) S_11_ parameter versus fat layer thickness; (**b**) gain (dB) versus fat layer thickness.

**Figure 8 bioengineering-10-00216-f008:**
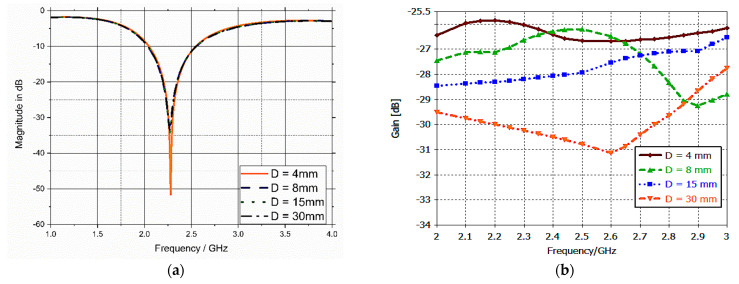
(**a**) S_11_ parameter versus implanting depth; (**b**) gain (in dB) versus implanting depth.

**Figure 9 bioengineering-10-00216-f009:**
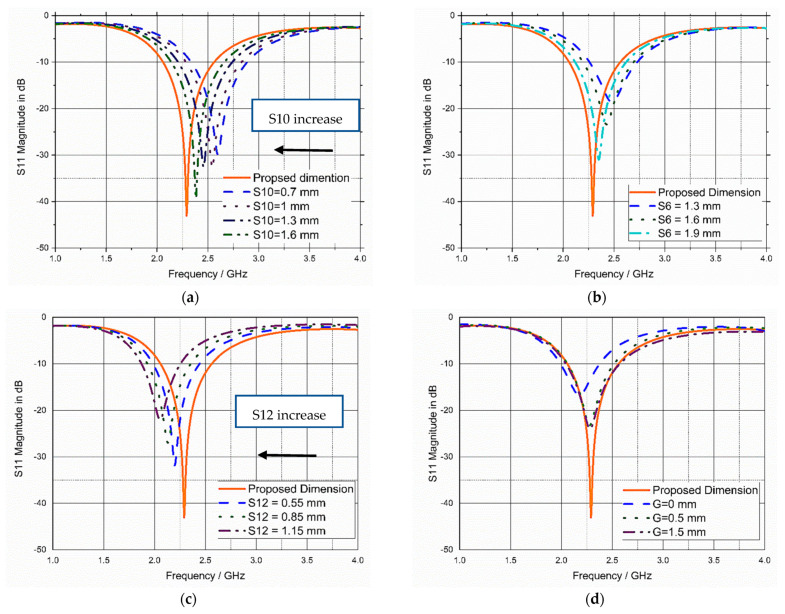
Impact of variation of (**a**) S10, (**b**) S6, (**c**) S12, and (**d**) G (partial ground length).

**Figure 10 bioengineering-10-00216-f010:**
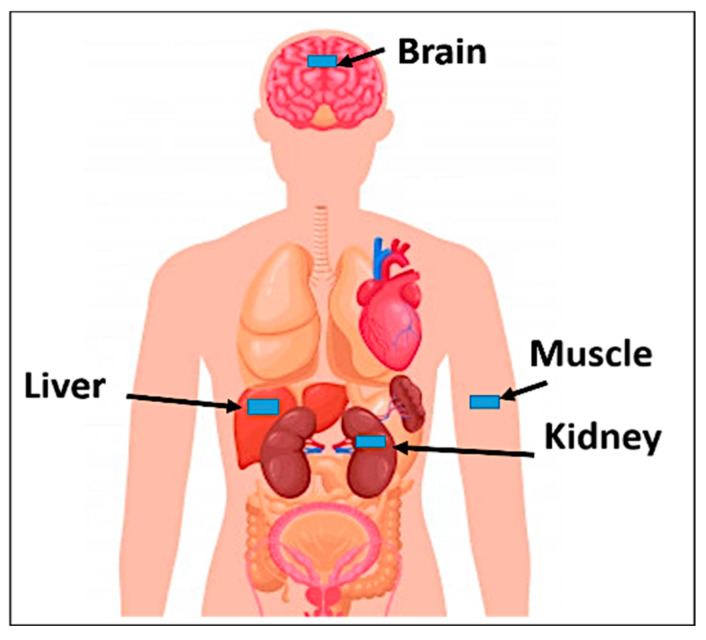
Distribution of areas where the antenna was implanted to be tested.

**Figure 11 bioengineering-10-00216-f011:**
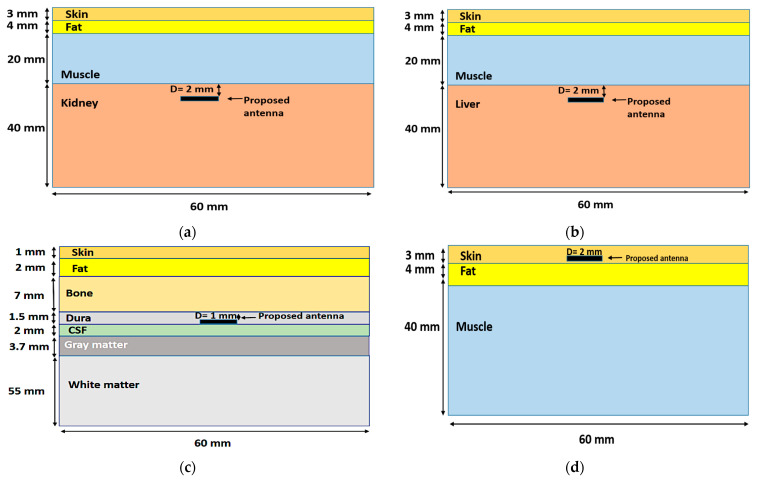
Realistic heterogeneous model mimicking the human body. (**a**) Four-layer kidney model; (**b**) four-layer liver model; (**c**) seven-layer brain model; (**d**) human body model.

**Figure 12 bioengineering-10-00216-f012:**
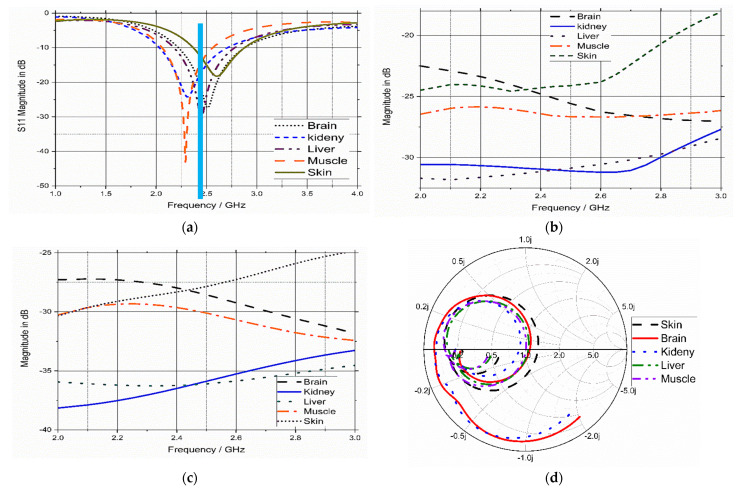
Simulated (**a**) S_11_ parameter, (**b**) gain, (**c**) total efficiency, and (**d**) input impedance of the proposed antenna.

**Figure 13 bioengineering-10-00216-f013:**
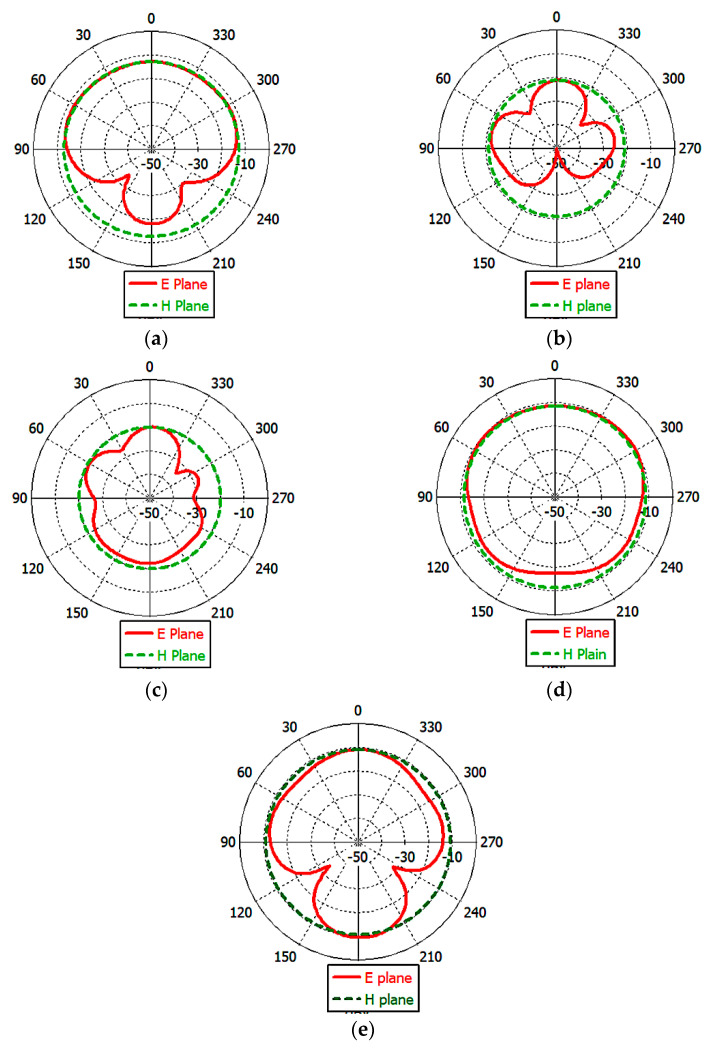
Radiation patterns of the proposed antenna at 2.45 GHz in the different scenarios: (**a**) muscle; (**b**) kidney; (**c**) liver; (**d**) brain; (**e**) skin.

**Figure 14 bioengineering-10-00216-f014:**
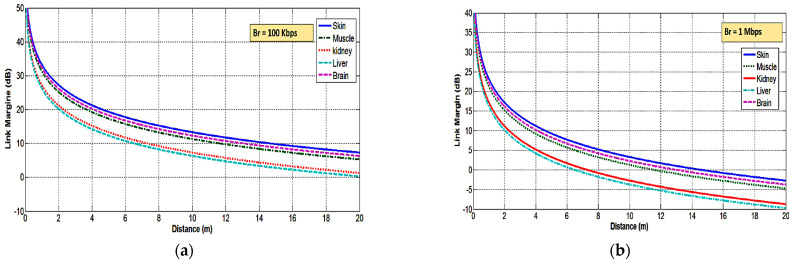
Calculated link margin at bit rate of (**a**) 100 Kbps and (**b**) 1 Mbps for different scenarios.

**Figure 15 bioengineering-10-00216-f015:**
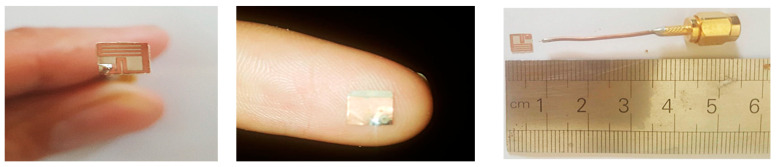
Photos of the prototypes fabricated antenna.

**Figure 16 bioengineering-10-00216-f016:**
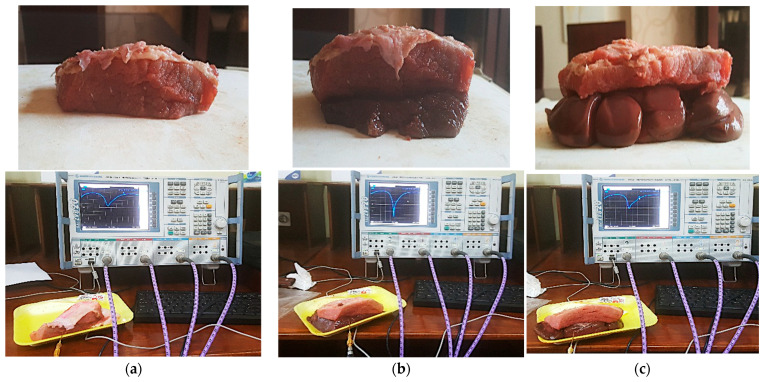
The measurement setup. (**a**) Muscle phantom; (**b**) Liver phantom; (**c**) Kidney phantom.

**Figure 17 bioengineering-10-00216-f017:**
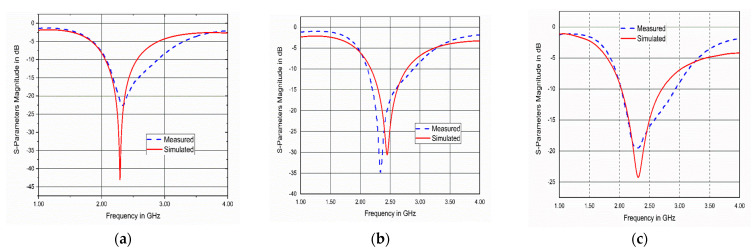
Simulated and measured reflection coefficients of the implanted antenna in different scenarios: (**a**) muscle; (**b**) liver; (**c**) kidney.

**Table 2 bioengineering-10-00216-t002:** The optimized parameters of the proposed WUCIA.

Parameters	Values (mm)	Parameters	Values (mm)
W	5	S6	2.2
L	5	S7	0.3
S1	4.2	S8	0.3
S2	0.3	S9	2.25
S3	1.5	S10	1.8
S4	1.15	S11	2.55
S5	0.3	G	1

**Table 3 bioengineering-10-00216-t003:** Max SAR and maximum allowed power for 1 g/10 g human tissue.

Human Tissue	Max. *SAR* (W/Kg)		Max. Allowed Input Power (mW)	
	1 g	10 g	1 g	10 g
Muscle	712.2	78.86	2.8	20.28
Kidney	771.4	82.98	2.59	19.28
Liver	758.2	83.4	2.64	19.18
Brain	788.7	85.24	2.535	18.77
Skin	715.7	77.6	2.79	20.6
Muscle	712.2	78.86	2.8	20.28

**Table 4 bioengineering-10-00216-t004:** Absorption rate calculation parameters.

Transmission		Receiver	
Frequency (GHz)	2.45	*R_x_* antennae gain *G_r_* (dBi)	2.15
*T_x_* power (dBm)	−40	*PLF* (dB)	1
*T_x_* antenna gain, *G_t_* (dBi)	Scenario dependent	Temperature *T_0_* (K)	293
Boltzmann constant (K)		1.38 × 10^−23^	
Signal quality			
Bit rate *B_r_* (Kb/s)		100/1000	
Bit error rate		1 × 10^−5^	
*E_b_/N_0_* (ideal PSK) (dB)		9.6	
Coding gain *G_c_* (dB)		0	

**Table 5 bioengineering-10-00216-t005:** Comparison of proposed antenna and similar prior studies in recent years.

Ref.	Freq. (GHz)	Volume mm^3^	BW (MHz)	Gain (dBi)	SAR W/kg	Implantation Scenario	Depth (mm)	Operation Condition
[56]	2.45	99.75	520	−26.4	712 (1 g)	skin	4	homogeneous
[57]	2.6	37	400	−19.7	0.7 (10 g)	skin	4	homogeneous
[58]	2.45	17.15	219	−18.2	305 (1 g)	scalp	3	homogeneous
[59]	1.42 and 2.42 GHz	91.4	140/240	−29.4 −21.2	500 (1 g) 686 (1 g)	skin	4	homogeneous
[60]	2.45	2.11	152.8	−24.5	233 (1 g)	muscle	50	homogeneous
Current work	2.45	6.5	520, 510, 656, 610, 664	−24, −26, −25, −31, −30	715.7 712.2 788.7 758.2 771.39	Skin, muscle, brain, liver, kidney	2 9 11 29 29	heterogeneous

## Data Availability

All the data have been included in the study.

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
