# Peer review of "Design and Modeling of Ultra-Compact Wideband Implantable Antenna for Wireless ISM Band"

_bioengineering, 2023, doi:10.3390/bioengineering10020216_

Round 1

Reviewer 1 Report

Authors present a wideband ultra-compact implantable antenna for wireless body area network (WBAN). The proposed patch antenna works in the industrial, scientific, and medical bands. The proposed patch antenna is characterized by wide bandwidth which makes the antenna unaffected by implantation in different human body parts. The SAR was calculated for health considerations. The result meet the requirements of the related standard (IEEE and ICNRP)

Tests provide good results. 

This paper is well written, I suggest the following points :

- In the introduction, The comparison with previous works must be more precise in order to highlight the real contribution of this work. Also, the main contribution should be more clarified.

- Title of subsection 2.1 is duplicated in 2.2 !!!

- Figure 3 : Please add details of each step in the figure title. 

-  English is generally good, but needs to be polished further. The manuscript should be formatted better and some spelling and grammar should be checked carefully.

- Conclusion should be rewritten integrating the limitations and the future works.

Concluding, the paper has potential to be appreciated by the readers and the above comment are formulated such that to enhance its impact.

Author Response

Dear reviewer 

Please find the attachment, which is the answer to your comments.

Best regards,

Reviewer 2 Report

Please ensure all tabulated data displays the proper amount of significant figures.

Was an established design procedure or protocol used for the design of this antenna? Please clarify.

Table 2: please include parameter units as appropriate.

Please indicate the antenna feed location in the design. It is not clear from the information presented where the antenna feed is located. The photos in figure 15 show this info but please include it in your design and simulation presentation.

Figures depicting model geometry: please consider using lighter colors, they can be difficult to read when printed in black and white. 

Figure 9: in figures where the response moves in a specific direction as you modify the variables it would be helpful to the reader to include arrows indicating the trending direction. 

Figure 17: are there any measures of the simulated and experimental data that could be used to compare the response with numbers? Maybe difference in center frequency or difference in bandwidth or Q-factor or some other standard measure to show the similarity or difference between the two. 

Overall well written and presented.

Author Response

(The authors gave the same response as above.)
